# Obatoclax and Paclitaxel Synergistically Induce Apoptosis and Overcome Paclitaxel Resistance in Urothelial Cancer Cells

**DOI:** 10.3390/cancers10120490

**Published:** 2018-12-05

**Authors:** Rocío Jiménez-Guerrero, Jessica Gasca, M. Luz Flores, Begoña Pérez-Valderrama, Cristina Tejera-Parrado, Rafael Medina, María Tortolero, Francisco Romero, Miguel A. Japón, Carmen Sáez

**Affiliations:** 1Instituto de Biomedicina de Sevilla (IBIS), Hospital Universitario Virgen del Rocío/CSIC/Universidad de Sevilla, 41013 Seville, Spain; rjimenez-ibis@us.es (R.J.-G.); jgasca-ibis@us.es (J.G.); mflores-ibis@us.es (M.L.F.); ctejera-ibis@us.es (C.T.-P.); mjapon@cica.es (M.A.J.); 2Department of Oncology, Hospital Universitario Virgen del Rocío, 41013 Seville, Spain; bperezv@gmail.com; 3Department of Urology, Hospital Universitario Virgen del Rocío, 41013 Seville, Spain; rantonio.medina.sspa@juntadeandalucia.es; 4Department of Microbiology, Faculty of Biology, Universidad de Sevilla, 41012 Seville, Spain; torto@us.es (M.T.); frport@us.es (F.R.); 5Department of Pathology, Hospital Universitario Virgen del Rocío, 41013 Seville, Spain

**Keywords:** bladder cancer, obatoclax, paclitaxel, autophagy, apoptosis

## Abstract

Paclitaxel is a treatment option for advanced or metastatic bladder cancer after the failure of first-line cisplatin and gemcitabine, although resistance limits its clinical benefits. Mcl-1 is an anti-apoptotic protein that promotes resistance to paclitaxel in different tumors. Obatoclax, a BH3 mimetic of the Bcl-2 family of proteins, antagonizes Mcl-1 and hence may reverse paclitaxel resistance in Mcl-1-overexpressing tumors. In this study, paclitaxel-sensitive 5637 and -resistant HT1197 bladder cancer cells were treated with paclitaxel, obatoclax, or combinations of both. Apoptosis, cell cycle, and autophagy were measured by Western blot, flow cytometry, and fluorescence microscopy. Moreover, Mcl-1 expression was analyzed by immunohistochemistry in bladder carcinoma tissues. Our results confirmed that paclitaxel alone induced Mcl-1 downregulation and apoptosis in 5637, but not in HT1197 cells; however, combinations of obatoclax and paclitaxel sensitized HT1197 cells to the treatment. In obatoclax-treated 5637 and obatoclax + paclitaxel-treated HT1197 cells, the blockade of the autophagic flux correlated with apoptosis and was associated with caspase-dependent cleavage of beclin-1. Obatoclax alone delayed the cell cycle in 5637, but not in HT1197 cells, whereas combinations of both retarded the cell cycle and reduced mitotic slippage. In conclusion, obatoclax sensitizes HT1197 cells to paclitaxel-induced apoptosis through the blockade of the autophagic flux and effects on the cell cycle. Furthermore, Mcl-1 is overexpressed in many invasive bladder carcinomas, and it is related to tumor progression, so Mcl-1 expression may be of predictive value in bladder cancer.

## 1. Introduction

Urinary bladder cancer is a major cause of morbidity and mortality worldwide, with over 400,000 new cases diagnosed per year [1]. Most cases are urothelial carcinomas, which often present as superficial papillary proliferations with a good overall survival rate. However, about thirty percent of cases present as muscle-invasive carcinomas, which carry a higher risk of lethal metastatic disease [2]. Muscle-invasive bladder carcinoma may need radical surgery and adjuvant chemotherapy with gemcitabine or platinum, and relapsed or refractory cases may require second-line therapies, which most often consist of chemotherapy with the microtubule toxins vinflunine, paclitaxel, or docetaxel [3]. These agents interfere with microtubule dynamics and activate the mitotic spindle assembly checkpoint to arrest cells in mitosis; thereafter, cells may die promptly in mitosis or escape post-mitotically towards other cell fates [4]. Paclitaxel and docetaxel are widely used in the treatment of various types of cancer, but resistance and toxicity often limit their clinical efficacy. Since spindle assembly checkpoint proteins are activated upon taxane treatment, they may be regarded as potential predictive markers of taxane resistance [5]. Anti-apoptotic proteins may also be involved in taxane resistance. We have previously shown that anti-apoptotic Mcl-1 is related to paclitaxel resistance in prostate and breast cancer and that Mcl-1 over-expression is a frequent event in aggressive cancers [6,7,8]. 

Obatoclax (GX15-070), a small molecule of the class of BH3 mimetics, antagonizes the anti-apoptotic Bcl-2 family of proteins by disrupting their interactions with Bax or Bak. This compound potently inhibits the association of Mcl-1 with Bak and is able to kill Mcl-1 overexpressing cancer cells, reverting Mcl-1-mediated resistance to Bcl-2/Bcl-xL/Bcl-w-selective antagonist ABT-737 [9,10]. Obatoclax has proven anti-tumor activity for several hematological malignancies when used as a single-agent therapy [11,12]. Based on its antagonistic role of the anti-apoptotic Bcl-2 family of proteins, obatoclax was first described to induce apoptotic cell death [9,13]. Other non-apoptotic forms of cell death, particularly autophagy, have also been demonstrated to play a role in obatoclax-mediated cytotoxicity, and several studies showed that the blockade of autophagy resulted in cell death since autophagy inhibitors potentiated the cytotoxicity of obatoclax [14,15,16,17,18]. Furthermore, it has been reported that obatoclax induces the alkalization and destabilization of lysosomes after the blockade of autophagic flux as a mechanism of cytotoxicity [19,20,21], and other cell death mechanism such as necroptosis have also been described [22,23].

There are many links between the pathways of apoptosis and autophagy. For example, anti-apoptotic protein Bcl-2 interacts with autophagy-related protein beclin-1 to inhibit beclin-1-dependent autophagy [24]. Bim, another Bcl-2 family member, can inhibit autophagy by recruiting beclin-1 to microtubules where Bim associates with dynein LC8 in the absence of apoptotic stimuli [25]. Furthermore, autophagy-related (Atg) proteins can be cleaved by cell death proteases calpain-1 and caspases-3, -6, and -8, providing the potential regulation of autophagy, as well as novel pro-apoptotic functions of cleaved fragments [26]. The caspase-mediated cleavage of beclin-1 was shown to yield fragments that lack the autophagy-inducing capacity of intact beclin-1, as well as a C-terminal fragment that localized predominantly in the mitochondria and sensitized cells to apoptosis [27].

In this study, we analyze the expression of Mcl-1 in a series of superficial and muscle-invasive bladder cancer patients and in paclitaxel-sensitive 5637 and paclitaxel-resistant HT1197 cell lines in response to paclitaxel and obatoclax treatment. We demonstrate that the Mcl-1 antagonist obatoclax, given in combination with paclitaxel, is able to sensitize resistant cells to apoptotic cell death. We investigate the effects of obatoclax and combinations of obatoclax and paclitaxel on the cell cycle and the crosstalk between apoptosis and autophagy. We show that in paclitaxel-resistant cancer cells, the combination of obatoclax with paclitaxel retards the cell cycle, inhibiting mitotic slippage and promoting the blockade of the autophagic flux to facilitate paclitaxel-induced apoptosis. 

## 2. Results

### 2.1. Decrease of Mcl-1 and Apoptosis Concur in Paclitaxel-Treated 5637, but Not in HT1197 Bladder Cancer Cells

We used 5637 and HT1197 bladder cancer cells to investigate the mechanisms of bladder cancer response to taxanes. Cells were treated with 0.1 µM paclitaxel for 24 and 48 h, and apoptosis was monitored by Western blot (Figure 1a). Apoptotic cell death was induced in 5637 cells, as shown by cleavage of PARP, caspase-9, and caspase-3, which were activated upon treatment. Paclitaxel in 5637 cells also decreased Mcl-1 levels and induced the phosphorylation of Bcl-xL, thus inhibiting the anti-apoptotic properties of this protein. Levels of pro-apoptotic Bax and Bak did not change after paclitaxel treatment. In contrast, paclitaxel did not induce apoptosis in HT1197 cells, as shown by the absence of cleavage of PARP, caspase-9, and caspase-3. Importantly, Mcl-1 levels did not drop in HT1197 cells, and Bcl-xL remained unphosphorylated, which suggests that these mechanisms may contribute to paclitaxel resistance. 

To check whether the bladder cancer cells were arrested in mitosis upon paclitaxel treatment or could escape by slippage, the levels of cyclin B1 and p-histone H3^Ser10^ were analyzed (Figure 1b). In paclitaxel-sensitive 5637 cells, the levels of cyclin B1 and p-histone H3^Ser10^ increased after 24 h and decreased after 48 h of paclitaxel treatment. In paclitaxel-resistant HT1197 cells, these mitotic proteins behaved similarly after paclitaxel treatment, with an increase in protein levels at 24 h and a decrease at 48 h. Thus, both cell lines are first arrested in mitosis and then exit mitosis by slippage. Analysis by FISH (Figure 1c) demonstrated that more than 80% of 5637 cells exhibited a duplicated DNA ploidy before late apoptosis as a result of mitotic slippage. Similarly, more than 90% of HT1197 cells had duplicated DNA ploidy upon paclitaxel treatment. These results indicate that 5637 and HT1197 cells share common mechanisms to escape paclitaxel-induced mitotic arrest, but only 5637 cells suffer apoptosis after mitotic slippage. 

Mcl-1 may have major roles in the distinct responses to paclitaxel in these bladder cancer cells and, therefore, is a good candidate to predict paclitaxel response in the clinical setting. Accordingly, we used immunohistochemistry to analyze the expression of Mcl-1 in biopsies from 72 bladder carcinomas (53 non-muscle-invasive, 19 muscle-invasive) (Figure 1d and Appendix A). Most non-muscle-invasive bladder carcinomas (96.2%) showed very low levels of Mcl-1. On the other hand, 12 out of 19 (63.2%) muscle-invasive bladder carcinomas showed increased levels of Mcl-1. Regarding the prognostic role of Mcl-1 expression, we performed a Kaplan–Meier study where we observed that high Mcl-1 levels were related to a lower disease free survival and a higher risk of patient recurrence, indicating that Mcl-1 could be a good prognostic marker of tumor aggressiveness (Figure 1e).

### 2.2. The Combination of Paclitaxel and Mcl-1 Antagonist Obatoclax Induces Apoptosis in Resistant HT1197 Cells

As mentioned, Mcl-1 may mediate paclitaxel resistance in bladder cancer cells and is highly expressed in many muscle-invasive bladder carcinomas. Obatoclax antagonizes Mcl-1 and induces cell death in hematological tumors [11,12], so we decided to study obatoclax effects in paclitaxel-sensitive and paclitaxel-resistant bladder cancer cells. We treated 5637 and HT1197 cells with 1 µM obatoclax either alone or in combination with 0.1 µM paclitaxel (Figure 2). In the latter, cells were treated with one drug for 8 h, and the other drug was added for 40 h or cells were treated with both drugs at the same time for 48 h. Paclitaxel-sensitive 5637 cells were responsive to obatoclax alone and in combination with paclitaxel, as shown by PARP cleavage and caspase-3 activation in all conditions. Moreover, Mcl-1 levels decreased after all of the treatments, while Bak levels showed no change. In the case of HT1197 cells, we observed no PARP cleavage, caspase-3 activation, or Mcl-1 decrease using either obatoclax or paclitaxel as single treatments. However, all three combinations of obatoclax and paclitaxel were able to induce apoptosis and also decrease Mcl-1 expression in HT1197 cells. Therefore, the combination of paclitaxel with obatoclax overcame the paclitaxel resistance of HT1197 cells.

### 2.3. LC3-II and p62 Accumulate in 5637 Cells Treated with Obatoclax and HT1197 Cells Treated with Combinations of Obatoclax and Paclitaxel

Cytotoxicity mediated by obatoclax has been associated with autophagy processes. We proceeded to study the autophagic flux in 5637 and HT1197 cells treated with obatoclax or combinations of obatoclax and paclitaxel and its relationship with cell death or survival. First, cells were treated with 1 µM obatoclax for 48 h in the presence or absence of 400 nM bafilomycin A1 (BafA1), an autophagic flux inhibitor, added for the last 4 h of treatment. Autophagic flux was monitored by Western blot of LC3-Band p62 (Figure 3a, Appendix A). In 5637 cells, obatoclax alone induced the accumulation of LC3-II and p62 more so than after the addition of BafA1. This accumulation reflects that obatoclax alone is able to cause an efficient blockade of the autophagic flux in 5637 cells. In contrast, obatoclax alone did not increase the protein levels of LC3-II and p62 in HT1197 cells as much as in 5637 or after the addition of BafA1, reflecting a less efficient blockade of the autophagic flux by obatoclax in HT1197 cells. Then, 5637 and HT1197 cells were treated with obatoclax, paclitaxel, and the combinations of obatoclax and paclitaxel, and the autophagic flux was studied by Western blot of LC3-Band p62 (Figure 3b, Appendix A). Paclitaxel alone did not increase the levels of LC3-II and p62 in either 5637 or HT1197 cells. All three combinations of obatoclax and paclitaxel induced the accumulation of LC3-II and p62 in 5637 cells and, interestingly, in HT1197 cells, in which the blockade of the autophagic flux may act as a mechanism of sensitization to paclitaxel.

### 2.4. Caspase-Dependent Cleavage of Beclin-1 Associates with Autophagy Blockade and Apoptosis in HT1197 Cells Treated with a Combination of Obatoclax and Paclitaxel

Beclin-1 shares a BH3-like domain with the anti-apoptotic members of the Bcl-2 family. Its interaction with Bcl-2 or Bcl-xL has inhibitory actions on autophagy, and its cleavage by caspase-3 regulates crosstalk between apoptosis and autophagy [24,27]. We analyzed the role of beclin-1 in the blockade of the autophagic flux and induction of apoptosis in cells treated with obatoclax or a combination of obatoclax and paclitaxel. First, 5637 cells were treated with 1 µM obatoclax in the presence or absence of caspase inhibitor Z-VAD-fmk, and beclin-1 was analyzed by Western blot (Figure 3c). Obatoclax treatment induced the cleavage of beclin-1, as shown by the appearance of 35- and 37-kDa fragments. Cleavage of beclin-1 was caspase-dependent, as fragments were not seen in the presence of caspase inhibitor Z-VAD-fmk. Then, we compared the cleavage of beclin-1 in 5637 and HT1197 cells treated with obatoclax alone or in combination with paclitaxel (Figure 3d, Appendix A). Again, cleavage of beclin-1 was observed in 5637 cells treated with obatoclax or with the combination of both drugs. In HT1197 cells, treatment with obatoclax alone did not induce cleavage of beclin-1. However, the combination of obatoclax and paclitaxel resulted in the appearance of 35- and 37-kDa fragments in the Western blot, reflecting that beclin-1 was cleaved and complete autophagy achieved. This mechanism may help to induce apoptosis when paclitaxel-resistant cells are treated with a combination of obatoclax and paclitaxel.

### 2.5. Blockade of the Autophagic Flux Correlates with Apoptotic Cell Death in HT1197 Cells Treated with a Combination of Obatoclax and Paclitaxel

To evaluate autophagic flux inhibition, cells were treated with 1 µM obatoclax alone or in combination with 0.1 µM paclitaxel for 48 h, with 800 nM rapamycin as the control for autophagic flux induction and 50 µM chloroquine as the control for autophagic flux inhibition (DMSO was used as the negative control). After treatment, cells were loaded with Cyto-ID^TM^ (Enzo Life Sciences, Farmingdale, NY, USA) Green dye and then analyzed by flow cytometry (Figure 4a). Using this method, fluorescence intensity correlates with the accumulation of autophagy vesicles as a result of inhibition of the autophagic flux. Mean fluorescence intensity for each condition was expressed as a fold change as compared to DMSO. In 5637 cells, obatoclax alone and the combination of obatoclax and paclitaxel caused a higher fluorescence intensity than that induced by chloroquine, reflecting a low autophagic flux. In HT1197 cells, obatoclax alone induced a similar fluorescence intensity to that induced by rapamycin, suggesting that high autophagic flux was turned on in these cells. The combination of obatoclax and paclitaxel was able to inhibit autophagic flux in HT1197 cells, as shown by a similar fluorescence intensity to that induced by chloroquine. Next, we used fluorescence microscopy to analyze the formation of different autophagy compartments since Cyto-ID^TM^ Green dye labels the autophagosomes with minimal staining of lysosomes or endosomes (Figure 4b). Cyto-ID^TM^ Green dye revealed the formation of autophagosomes that do not fuse with lysosomes by means of perinuclear or cytoplasmic puncta in 5637 cells treated with obatoclax alone or in combination with paclitaxel. Counterstaining with Hoechst 33342 dye revealed nuclear fragmentation and the formation of apoptotic bodies in 5637 cells under both treatments. In HT1197 cells, obatoclax alone did not form Cyto-ID^TM^ Green dye-labeled puncta, but rather dark autophagolysosomes, indicating that autophagy was complete. Nuclear staining with Hoechst 33342 showed no apoptotic damage. However, the combination of obatoclax and paclitaxel was able to induce the accumulation of autophagosomes, labeled by the Cyto-ID^TM^ Green dye, and the apoptotic fragmentation of nuclei in HT1197 cells, reflecting that sensitization with obatoclax blocks the autophagic flux and facilitates paclitaxel-induced apoptosis in these cells.

### 2.6. Obatoclax Retards the Cell Cycle of Paclitaxel-Treated HT1197 Cells and Favors Inhibition of Mitotic Slippage

We studied the effects of obatoclax and the combinations of obatoclax and paclitaxel on the cell cycle of 5637 and HT1197 cells. DNA content was analyzed by flow cytometry 24 h and 48 h after the different treatment combinations (Figure 5a). Obatoclax alone increased the S-phase cell count at 48 h as compared with 24 h in 5637 cells, but not in HT1197 cells, which showed similar S-phase cell counts at 24 h and 48 h. Paclitaxel alone induced G2/M arrest at 24 h in both 5637 and HT1197 cells that persisted at 48 h, although this peak may be enriched in cells with higher ploidy. All three combinations of paclitaxel and obatoclax arrested 5637 cells in G2/M at 48 h, but the highest S-phase cell counts at 24 h where observed when obatoclax was added first. In contrast, the same combinations retarded the cell cycle in HT1197 cells, as shown by a higher G1-phase percentage compared with 5637 cells. As mentioned, paclitaxel treatment induces mitotic slippage in both 5637 and HT1197 cell lines, but with different end effects, i.e., cell death or survival, respectively. We performed ploidy analysis by FISH to study the effects of the combination of paclitaxel and obatoclax on mitotic slippage (Figure 5b). Obatoclax alone did not alter ploidy in 5637 cells, as opposed to paclitaxel, which increased ploidy in these cells. The combination of paclitaxel and obatoclax in 5637 cells had different effects depending on the treatment sequence: obatoclax followed by paclitaxel had a similar effect on ploidy as obatoclax alone, whereas paclitaxel followed by obatoclax or both drugs added together increased ploidy in 5637 cells. This means that obatoclax alone or when added first may induce cell death in 5637 cells before mitosis, but when paclitaxel is added first or together with obatoclax, mitotic slippage is the principal means of cell death. In HT1197 cells, obatoclax alone did not change ploidy, as compared with DMSO. Paclitaxel alone increased ploidy of HT1197 cells, a sign of mitotic slippage as a mechanism of resistance in these cells. None of the three combinations of paclitaxel and obatoclax changed ploidy in HT1197 cells, reflecting that paclitaxel-induced G2/M arrest was more efficient and mitotic slippage restrained. 

### 2.7. Combinations of Obatoclax and Paclitaxel Promote Apoptosis in Paclitaxel-Resistant HT1376 and MDA-MB-231 Cells

To confirm these results, we analyzed two further cancer cell lines, HT1376 (bladder cancer) and MDA-MB-231 (breast cancer) cells, that are resistant to paclitaxel, but, in contrast with HT1197, more sensitive to obatoclax as a single treatment. These cells were treated with 1 µM obatoclax, 0.1 µM paclitaxel, or combinations, as described. In both cell types, all combinations of obatoclax and paclitaxel were able to induce higher levels of apoptosis than obatoclax or paclitaxel alone, as shown by cleavage of PARP and caspase-3, besides decreasing Mcl-1 levels (Figure 6a, Appendix A). We checked the autophagic flux by Western blot of LC3-B and p62, and all combinations of obatoclax and paclitaxel were able to maintain the blockade of the autophagic flux, as seen with obatoclax alone in both cell lines (Figure 6a, Appendix A).

Cell cycle analysis of HT1376 cells treated with obatoclax alone showed no dramatic effects with respect to DMSO at 24 h and 48 h as previously observed for HT1197; however, in MDA-MB-231 breast cancer cells, obatoclax alone clearly increased S-phase. Paclitaxel alone induced G2/M arrest at 24 h, which persisted at 48 h in both cell types. Combinations of obatoclax and paclitaxel increased S-phase with respect to paclitaxel alone, particularly when obatoclax was added first (Figure 6b, Appendix A). FISH analysis showed that obatoclax alone did not change the ploidy of HT1376 or MDA-MB-231 cells. In contrast, paclitaxel increased the ploidy as a sign of mitotic slippage in both cell types. Combinations of obatoclax and paclitaxel were able to reduce the number of cells with higher ploidy with respect to paclitaxel alone, reflecting that mitotic slippage inhibition may also account for apoptotic cell death.

## 3. Discussion

Paclitaxel is used in the second-line therapy of advanced urothelial cancer, but resistance and toxicity limit its clinical benefits. Elucidation of the mechanisms of paclitaxel resistance may help to identify biomarkers for the development of alternative or sensitizing therapies. We previously showed that Mcl-1 is a determinant of the response to paclitaxel treatment, is overexpressed in many paclitaxel-resistant tumors, and that downregulation of Mcl-1 restores the sensitivity to paclitaxel [6,7,8]. Herein, we show that Mcl-1 is downregulated in sensitive 5637 cells, but not in resistant HT1197 cells in response to paclitaxel and that Mcl-1 is overexpressed in many muscle-invasive bladder carcinomas. The relevance of Mcl-1 as a biomarker of aggressiveness is well described in several malignancies, and its overexpression has also been related to the resistance of several BH3 mimetics [28,29,30]. We found that combinations of obatoclax and paclitaxel were able to induce apoptotic cell death in resistant HT1197 cells, as well as in other models of paclitaxel resistance such as HT1376 and MDA-MB-231 cells. Other authors have studied the role of obatoclax in paclitaxel sensitization in several cells lines. Stamelos et al. used a combination of both drugs and did not observe any sensitization to paclitaxel in ovarian cancer cells, although obatoclax sensitized cells to carboplatin treatment [20]. In hepatoblastoma cells, the combination of obatoclax with paclitaxel was reported to have additive effects, with a loss of cell viability [31].

Although several mechanisms have been implicated in the induction of obatoclax cell death—apoptosis, necroptosis, and autophagy—they are still not completely understood [14,15,16,19,32]. Several studies point to the blockade of autophagy as a mechanism of obatoclax toxicity due to lysosome impairment. In this sense, Champa et al. reported that short obatoclax treatment induces necrotic cell death in thyroid cancer cells by the destabilization of lysosomes after the blockade of the autophagic flux [19], and Stamelos et al. also found that obatoclax accumulates in lysosomes, inducing their alkalization and impairment in ovarian cancer cells, but without clear apoptosis induction [20], thus arguing that there must be other mechanisms of cell death other than apoptosis in ovarian cancer cells after obatoclax treatment. Moreover, in esophageal cancer cells, obatoclax has been reported to block autophagy by the impairment of the lysosome, inducing a cytotoxic effect with diminished cell viability [21]. Deregulation of autophagy has been implicated in many diseases, although the relationship between autophagy and apoptosis is still a matter of controversy. However, several studies indicate that these two processes are related and are important to the chemotherapy response [27,33]. Our results clearly show that obatoclax induces autophagy in bladder cancer cells and that this may be accompanied by apoptotic cell death after the blockade of the autophagic flux, which could contribute to the paclitaxel sensitization in paclitaxel-resistant cells. However, these processes are opposite and do not co-occur, since we just observe apoptosis when the autophagy is blocked. It is important to note that when autophagy is completed, as observed in the HT1197 cells as a mechanism of resistance, obatoclax does not induce cell death. This observation is in agreement with several studies that have shown that autophagy plays a protective role enhancing cellular survival in response to stress [34] and drug-resistance after chemotherapy treatment [35,36]. This “dual” role may be due to the difference in the autophagic flux, since it is not only important whether autophagy is occurring, but also whether the induced autophagy is completed. Thus, in agreement with our results observed for the obatoclax plus paclitaxel treatment of paclitaxel-resistant HT1197, HT1376, and MBA-MB-231 cancer cells, the autophagy blockade increases the therapeutic efficacy of chemotherapy in several malignancies [37,38] and induces apoptosis [39].

Beclin-1 is a molecule with a role in the initiation of autophagy after nutrient starvation and is also a caspase substrate, connecting autophagy with Bcl-2 family members via its BH3 domain [40]. Indeed, Wirawan et al. reported a novel pro-apoptotic function of beclin-1, demonstrating that the cleavage of becline-1 by caspase-3 results in the generation of a fragment that induces the inhibition of autophagy and the amplification of apoptosis via the release of cytochrome C from the mitochondria [27]. Other authors reported that the specific blockage of beclin-1 cleavage induces autophagy in cells treated with caspase inhibitors, indicating that beclin-1 cleavage by caspases is sufficient to suppress autophagy [41]. In this work, we observed that obatoclax treatment induced apoptotic cell death in the 5637 bladder cell line, and this was accompanied by beclin-1 cleavage and blockade of the autophagic flux. Furthermore, we demonstrated that the HT1197 bladder cancer cell line is resistant to obatoclax treatment when autophagy is complete; however, when we used obatoclax in combination with paclitaxel, apoptotic cell death was induced after the autophagy blockade and beclin-1 cleavage. Furthermore, the Mcl-1/Beclin-1 interaction has a role in the induction of autophagy and the survival of cells under stress conditions [40]. This phenomenon could induce the disruption of the Mcl-1/beclin-1 interaction that promotes Mcl-1 degradation and beclin-1 cleavage in a caspase-3-dependent manner and could be responsible for the apoptosis observed after the combination treatment.

Finally, we demonstrated that obatoclax treatment increased the S-phase in 5637 bladder and MDA-MB 231 breast cancer cells (but did not alter the cell cycle in HT1197 bladder cancer cells) and related the increment of the S-phase to apoptotic cell death. Our results differ from previous studies with colorectal [42] and esophageal [43] cancer cells, where a growth inhibitory dose of obatoclax caused arrest in G1 with some delay in the cell cycle progression not related to apoptotic cell death. Furthermore, the combinations of obatoclax and paclitaxel were not only able to retard the cell cycle progression in paclitaxel-resistant HT1197cells, but also to prevent the mitotic slippage that occurs when these cells are treated with paclitaxel alone. In these conditions, Mcl-1 was downregulated in HT1197 cells treated with combinations of obatoclax and paclitaxel, and we postulate that the cell cycle arrest facilitates the degradation of Mcl-1. Previous reports highlighted the importance of Mcl-1 ubiquitination and destruction by the proteasome for the promotion of apoptosis during the prolonged mitotic arrest induced by microtubule toxins [8,44,45]. We suggest that the inhibition of mitotic slippage and the delay of cell cycle progression observed after combination treatment in HT1197 cells could increase the degradation of Mcl-1 by some ubiquitin ligases and thus influence the stability of Mcl-1 in mitosis, helping cell sensitization to paclitaxel. The same results were seen in two more paclitaxel-resistant bladder and breast cancer cell lines, so we can conclude that obatoclax helps to overcome paclitaxel sensitization by arresting cells in the S/G2-phase to escape mitotic slippage as a mechanism of resistance.

## 4. Materials and Methods

### 4.1. Cell Culture and Drugs

Human 5637 and HT1197 bladder cancer cells and MDA-MB-231 breast cancer cells were ordered from the Interlab Cell Line Collection (Genoa, Italy), and HT1376 bladder cancer cells were from Sigma (St. Louis, MO, USA). All experiments were performed using cells that had not exceeded the first ten passages after receipt of the initial vial and were routinely tested for *Mycoplasma* contamination. Cells were cultured in RPMI-1640 (Lonza, Basel, Switzerland) supplemented with 10% fetal bovine serum (Biochrom, Cambridge, UK), 50 U/mL penicillin and 50 mM streptomycin (Sigma), 10 mM HEPES (Lonza) and 1 mM glutamine (Gibco, Thermo Fisher Scientific, Waltham, Massachusetts, USA) at 37 °C in a humidified incubator under 5% CO_2_. The stock solutions of paclitaxel (Calbiochem, San Diego, CA, USA) and obatoclax (Selleck, Houston, TX, USA) were prepared at 10 mM in dimethyl sulfoxide (DMSO, Sigma) and stored at −20 °C. In all experiments, cells were treated with either drug or vehicle during the log phase of growth. Cells were treated with 1 µM obatoclax or 0.1 µM paclitaxel either as single treatment for 48 h or in combination: one drug for 8 h; and then, the other drug was added for 40 h or both drugs were added simultaneously for 48 h. The stock solutions of bafilomycin A1 and z-VAD-fmk (Selleck) were prepared at 10 mM in DMSO, and rapamycin and chloroquine (Enzo Life Sciences) were prepared at 60 mM and 500 µM, respectively, and stored at −20 °C.

### 4.2. Antibodies

Mouse monoclonal anti-PARP (1:500), anti-beclin-1 (1:500), rabbit polyclonal anti-Bax (1:2000), and anti-Bak (1:3000) were from BD Biosciences (San Jose, CA, USA); mouse monoclonal anti-Bcl-xL (1:1000), rabbit polyclonal anti-Mcl-1 (1:1000), anti-cyclin B1 (1:500), and anti-p-histone H3 (Ser10) (1:1000) were from Santa Cruz (Santa Cruz, CA, USA); mouse monoclonal anti-β-actin (1:10,000), rabbit polyclonal anti-LC3B (1:2000), and anti-p62 (1:2000) were from Sigma; rabbit polyclonal anti-cleaved caspase-9 (Asp315) (1:500) and anti-cleaved caspase-3 (Asp175) (1:500) were from Cell Signaling (Danvers, MA, USA).

### 4.3. Western Blot

Cells were lysed in Nonidet P-40 (NP40) lysis buffer (10 mM Tris-HCl (pH 7.5), 150 mM NaCl, 10% glycerol, and 1% NP40). Equal amounts of total protein, as determined by the BCA protein assay kit (Pierce, Rockford, IL, USA), were separated by SDS-PAGE on 8% polyacrylamide gels and transferred to Hybond ECL nitrocellulose membranes (GE Healthcare, Europe GmbH, Freiburg, Germany). Blots were stained with Ponceau S to ensure protein amounts were equal. For immunodetection, blots were soaked in 1% blocking reagent (Roche, Basel, Switzerland) in 0.05% Tween 20-PBS for 1 h and incubated with primary antibody in blocking buffer overnight at 4 °C. Blots were then washed in 0.05% Tween 20-PBS and incubated with either goat anti-mouse IgG (1:20,000; GE Healthcare) or goat anti-rabbit IgG (1:20,000; GE Healthcare) peroxidase-labeled antibodies in blocking buffer for 1 h. An enhanced chemiluminescent ECL system (GE Healthcare) was applied according to the manufacturer’s protocol. The experiments were performed in triplicate. Scanning densitometry of blots was analyzed using ImageJ software (Rasband, W.S., US National Institutes of Health, Bethesda, MD, USA, http://imagej.nih.gov/ij/). 

### 4.4. Flow Cytometric Analysis of Cell Cycle

Cells were trypsinized and fixed in 70% ethanol. Propidium iodide staining of nuclei was performed with the CycleTest Plus DNA reagent kit (BD Biosciences). DNA content was measured using CellQuest Pro software in a FACScan flow cytometer (BD Biosciences).

### 4.5. Fluorescence In Situ Hybridization

Cells were imprinted onto silanized slides and fixed in ice-cold methanol/glacial acetic acid (3:1). Slides were immersed in a 2× SSC (Saline Sodium Citrate)/0.3% NP40 solution at 37 °C during 30 min and then dehydrated. Cellular DNA and the Spectrum green-labeled chromosome 17 centromeric probe (Vysis) were co-denatured at 72 °C for 5 min and hybridized at 37 °C overnight. Slides were washed in 2× SSC/0.3% NP40 at 72 °C for 5 min, counterstained with DAPI, and visualized using a fluorescence microscope (Leica, Wetzlar, Germany). At least 100 cells were counted to calculate the percentage of cells with normal ploidy and higher ploidy in each condition.

### 4.6. Immunohistochemistry

Formalin-fixed, paraffin-embedded tissues from the transurethral resections of 72 patients with bladder carcinoma were selected to make tissue microarrays with 1 mm cores in duplicate. The study was approved by the local ethical committee. Five-micrometer tissue sections were dewaxed, rehydrated, and immersed in 3% H_2_O_2_ aqueous solution for 30 min to exhaust endogenous peroxidase. Heat-induced epitope retrieval was performed with 1 mM EDTA (pH 9.0) in a microwave oven. Sections were incubated overnight at 4 °C with anti-Mcl-1 antibody (1:1500). Peroxidase-labeled secondary antibody and 3,3′-diaminobenzidine were applied according to the manufacturer’s protocol (Dako, Glostrup, Denmark). Slides were then counterstained with hematoxylin and mounted. Immunostains were scored as low (nil or < 10% positive cells) or high Mcl-1 expression (≥ 10% positive cells).

### 4.7. Fluorescent Autophagy Detection Assay

The Cyto-ID^TM^ Autophagy Detection kit (Enzo) was used in order to detect the autophagic flux. Cells were treated with 1 µM obatoclax or 0.1 µM paclitaxel and 1 µM obatoclax for 48 h. As positive controls, to block or induce autophagic flux, cells were incubated with 50 µM chloroquine or 800 nM rapamycin for 18 h, respectively. The cells were then stained with Cyto-ID^TM^ Green dye and Hoechst 33342 using the Cyto-ID^TM^ Autophagy Detection Kit (Enzo) according to the manufacturer’s protocol, visualized using an inverted fluorescence microscope (Zeiss, Oberkochen, Germany) and counted using CellQuest Pro software in a FACScan flow cytometer (BD Biosciences).

### 4.8. Statistical Analysis

Data comparing differences between two conditions were statistically analyzed, when indicated, using the paired Student’s *t*-test. The association between the expression of Mcl-1 and patients’ tumor infiltration was analyzed using Fisher’s exact test. Recurrence-free survival curve was calculated by the method of Kaplan and Meier. The comparison of the survival curve was done by the log rank test of Mantel and Cox. Differences were considered significant when *p* < 0.05. Calculations were performed using Prism 6.0 (GraphPad, San Diego, CA, USA).

## 5. Conclusions

In summary, in this work, we explored the mechanism by which bladder cancer cells can overcome resistance to paclitaxel by combination treatment with obatoclax. Our results implicate the blockade of autophagy by the cleavage of beclin-1 as obatoclax-induced mechanisms of sensitization to apoptotic cell death. Moreover, the inhibition of mitotic slippage observed in paclitaxel-resistant cells (HT1197, HT1376, and MDA-MB-231) after the combination treatment could represent another mechanism to overcome paclitaxel resistance. Finally, Mcl-1 expression may be a useful predictive biomarker in advanced bladder cancer.

## Figures and Tables

**Figure 1 cancers-10-00490-f001:**
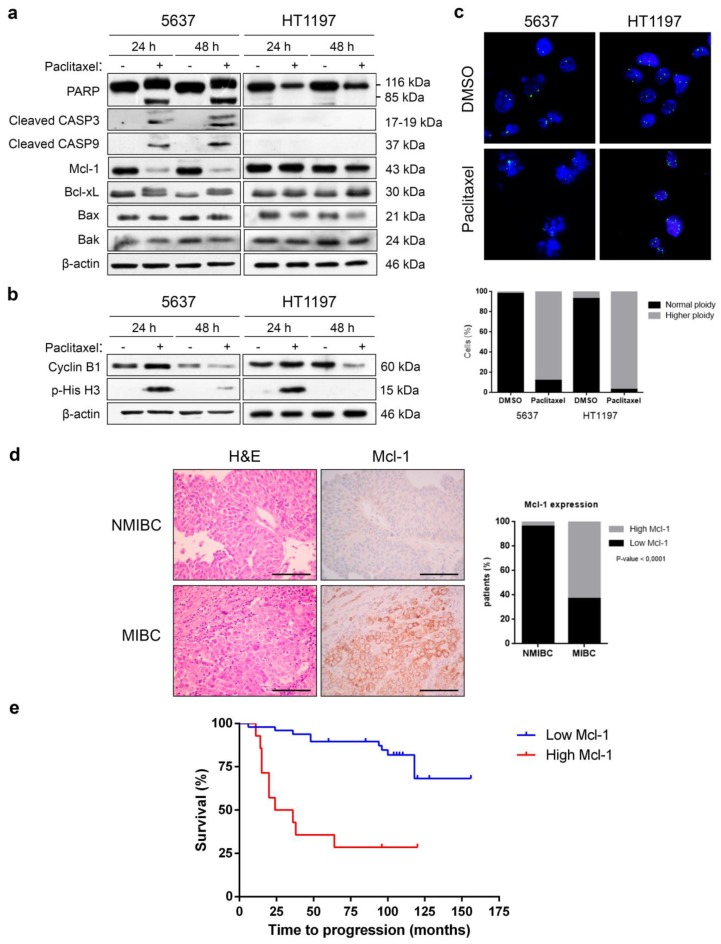
Downregulation of Mcl-1 is necessary for paclitaxel-induced apoptosis of bladder cancer cells. (**a**,**b**) 5637 and HT1197 cells were treated with dimethyl sulfoxide (DMSO) or 0.1 µM paclitaxel for 24 and 48 h. Western blot analyses of PARP, cleaved caspase-3, cleaved caspase-9, Mcl-1, Bcl-xL, Bax, Bak, cyclin B1, and p-histone H3^Ser10^ are shown. β-actin was used as a loading control. (**c**) Cells were subjected to FISH with a centromeric probe specific for chromosome 17 (Spectrum green). DNA was stained with DAPI (blue). At least 100 cells were counted for each condition, and the percentage of cells with normal ploidy or higher ploidy is presented in the histogram. Representative images are shown. Photomicrographs were taken using a 40× objective. (**d**) The expression of Mcl-1 was evaluated in samples of 72 patients with muscle-invasive or non-muscle-invasive bladder carcinoma by immunohistochemistry and quantified in the histogram. The relationship between the levels of expression of Mcl-1 and tumor infiltration in patients with bladder carcinoma was statistically significant. *p*-values were obtained using Fisher’s exact test. NMIBC: non-muscle-invasive bladder cancer; MIBC: muscle-invasive bladder cancer. Bars represent 100 μm. (**e**) Kaplan–Meier analysis of recurrence-free survival in patients with low (blue) and high (red) Mcl-1 expression. Tick marks represent censored patients. *p*-value < 0.0001 was obtained using the log-rank test of Mantel and Cox.

**Figure 2 cancers-10-00490-f002:**
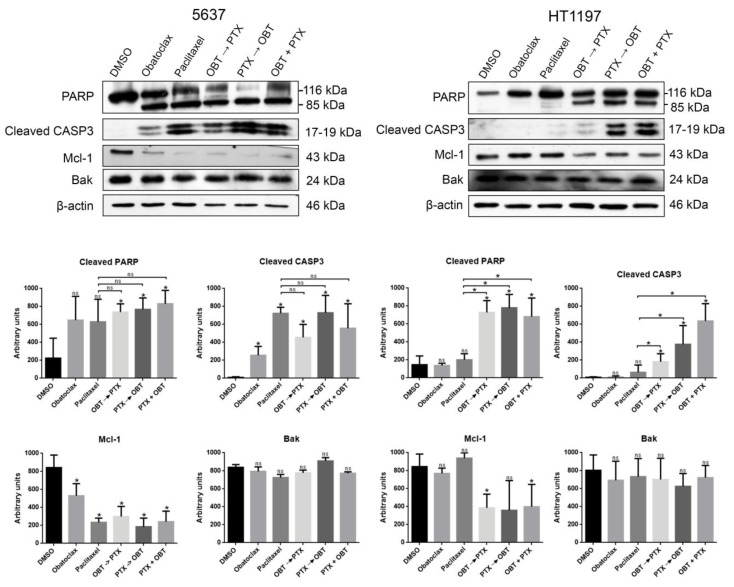
The combined treatment of Mcl-1 antagonist obatoclax with paclitaxel induces apoptosis in HT1197 cells. Cells were treated with 1 μM obatoclax, 0.1 µM paclitaxel, and combinations of both for 48 h. The combinations were: obatoclax followed by paclitaxel after 8 h, paclitaxel followed by obatoclax after 8 h, and both at the same time. In the 5637 and HT1197 cell lines, Western blot analyses of PARP, cleaved caspase 3, Mcl-1, and Bak are shown. β-actin is shown as a loading control. Histograms show the densitometric analysis of indicated proteins. Data are presented as the mean ± SD, and each treatment was compared with DMSO as the control. * *p*-value < 0.05 and ns indicates non-significant differences from Student’s *t*-test (n ≥ 3). OBT: obatoclax; PTX: paclitaxel.

**Figure 3 cancers-10-00490-f003:**
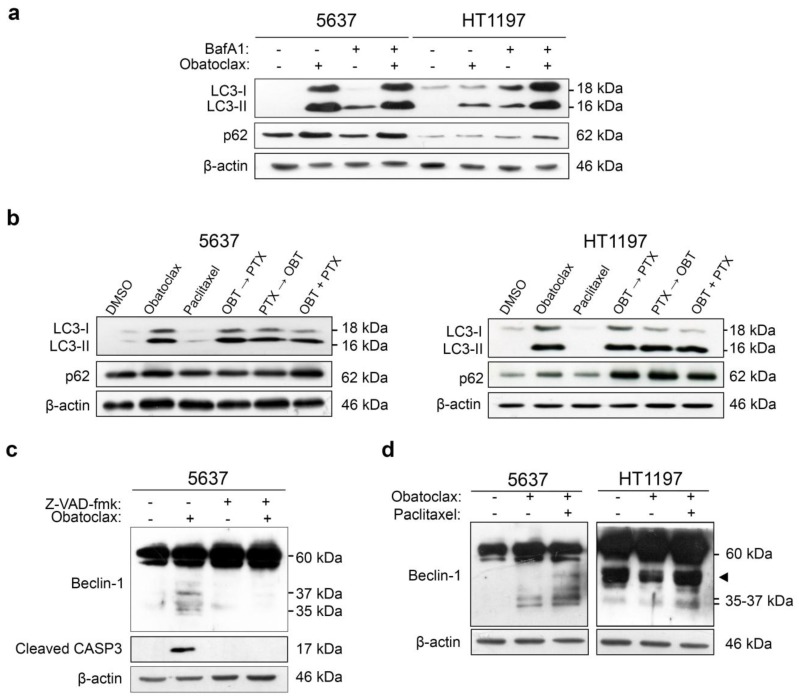
LC3-II and p62 accumulate in obatoclax-treated 5637 cells and HT1197 cells treated with obatoclax plus paclitaxel. The cleavage of beclin-1 by caspase 3 induces apoptosis as a mechanism of sensitization. (**a**) Cells were treated with DMSO or 1 µM obatoclax for 48 h, in the absence or presence of the autophagy inhibitor bafilomycin A1 400 nM for 4 h. (**b**) Cells were treated with 1 μM obatoclax, 0.1 µM paclitaxel, and combinations of both for 48 h. Western blot analyses of LC3-B and p62 are shown. β-actin is shown as a loading control. OBT: obatoclax; PTX: paclitaxel. (**c**) The 5637 cell line was treated with DMSO or 1 µM obatoclax for 48 h in the presence or absence of pre-incubation with pan-caspase inhibitor Z-VAD-fmk 20 µM for 1 h. (**d**) 5637 and HT1197 cells were treated with DMSO, 1 μM obatoclax, or 1 µM obatoclax plus 0.1 µM paclitaxel for 48 h. Western blot analyses of beclin-1 and cleaved caspase-3 are shown. Symbol ◄ indicates an unspecific band of 50-kDa beclin-1. β-actin is shown as a loading control.

**Figure 4 cancers-10-00490-f004:**
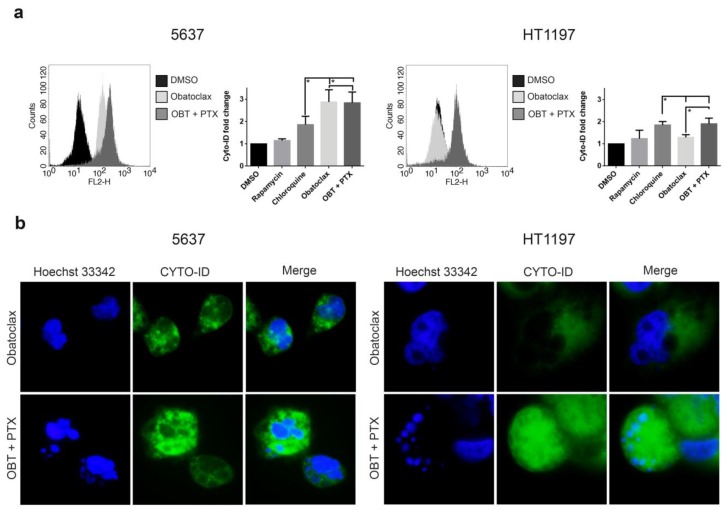
Autophagic flux blockade correlates with apoptotic cell death in obatoclax plus paclitaxel-treated HT1197 cells. Cells were treated with 1 µM obatoclax or 1 µM obatoclax and 0.1 µM paclitaxel for 48 h. DMSO, 800 nM rapamycin, and 50 µM chloroquine were used as controls. (**a**) Obatoclax-blocked autophagic flux at 48 h was measured by flow cytometry-based profiling of Cyto-ID^TM^ Autophagy Detection in 5637 and HT1197 cell lines. Histograms show the mean intensity of each treatment. Data were presented as the mean ± SD. * *p*-value < 0.05 from Student’s *t*-test (n ≥ 3). (**b**) Colocalization of the Cyto-ID^TM^ fluorescent dye and Hoechst 33342 in 5637 and HT1197 cells. Photomicrographs were taken using a 100× objective under an inverted fluorescence microscope. OBT: obatoclax; PTX: paclitaxel.

**Figure 5 cancers-10-00490-f005:**
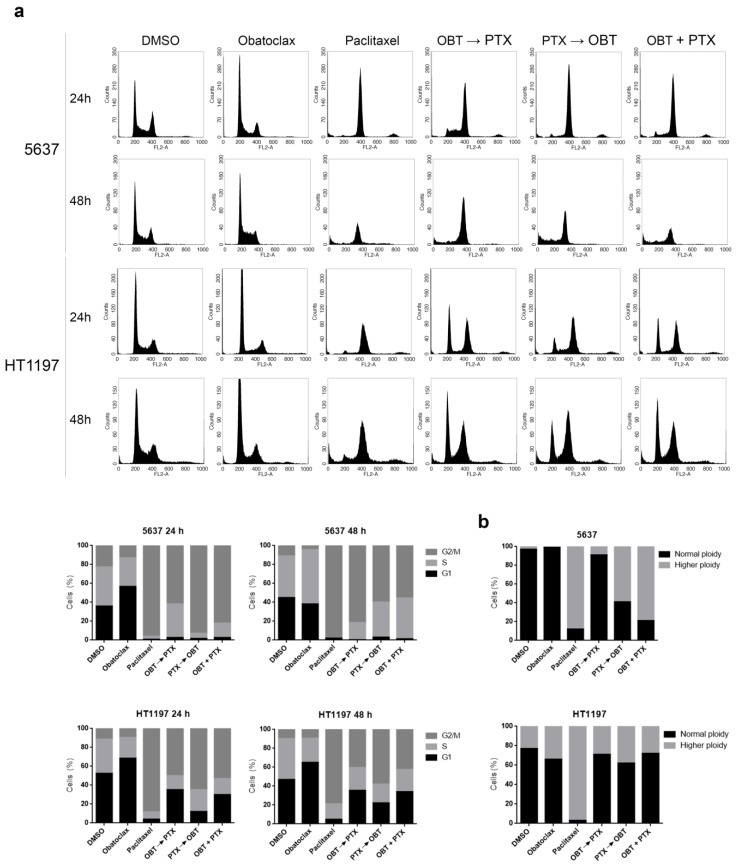
Obatoclax induces S arrest and a delayed cell cycle progression when combined with paclitaxel. Cells were treated with 1 μM obatoclax, 0.1 µM paclitaxel, and combinations of both for 48 h. (**a**) Cell cycle analysis of propidium iodide-stained cells by flow cytometry. Quantification of each phase is shown in the histograms. (**b**) Ploidy analysis for chromosome 17 by FISH. At least 100 cells were counted for each condition, and the percentage of cells with normal ploidy or higher ploidy is presented in the histogram. OBT: obatoclax; PTX: paclitaxel.

**Figure 6 cancers-10-00490-f006:**
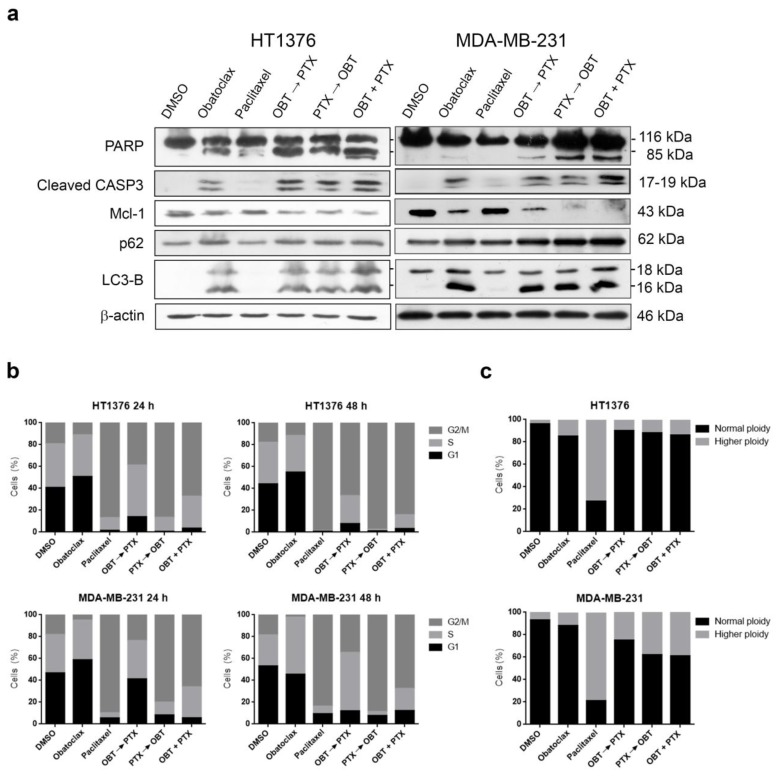
The combination of paclitaxel and obatoclax in HT1376 and MDA-MB-231 cells overcomes paclitaxel resistance. HT1376 and MDA-MB-231 cells were treated with 1 μM obatoclax, 0.1 µM paclitaxel, and combinations of both for 48 h. (**a**) Western blot analyses of PARP, cleaved caspase-3, Mcl-1, p62, and LC3-B are shown. β-actin was used as a loading control. (**b**) Cell cycle analysis of propidium iodide-stained cells by flow cytometry. The quantification of each phase is shown in the histograms. (**c**) Cells were subjected to FISH with a centromeric probe specific for chromosome 17 (Spectrum green). DNA was stained with DAPI (blue). At least 100 cells were counted for each condition, and the percentage of cells with normal ploidy or higher ploidy is presented in the histogram.

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
