# Peer review of "Obatoclax and Paclitaxel Synergistically Induce Apoptosis and Overcome Paclitaxel Resistance in Urothelial Cancer Cells"

_cancers, 2018, doi:10.3390/cancers10120490_

Reviewer 1 Report

The authors present a well-written article. A few concerns noted:

1) In Figure 1D, the IHC picture for Mcl-1 in NMIBC is not very clear, and can be improved.

2) The authors should show cell survival or proliferation data for treatments with obatoclax or paclitaxel.

3) The effects of obatoclax on normal urothelial cells should be investigated.

4) The authors should discuss whether autophagy and apoptosis co-occur in the cells treated with obatoclax.

Author Response

We gratefully acknowledge the positive general comments made by the reviewers and their critiques and informed suggestions, which we believe have given the manuscript more clarity and strength. An explanation of the changes that have been made is given below in response to each of the reviewer’s comments. We believe our revisions have addressed the reviewer’s concerns adequately. We also thank the Editor for her work and the opportunity given to us to submitting this revised version of the manuscript. 

We attach the reviewer's comments in a PDF file. 

Reviewer 2 Report

The authors evaluated the biological role of Mcl-1 in chemo-resistance of paclitaxel using two different cell lines of 5637 and HT1197 bladder cancer cell lines. They found that obatoclax, which is an antagonist of Mcl-1 could enhance the cytotoxic effect of paclitaxel in paclitaxel-resistant HT1197 cells through blocking autophagic flux, inducing apoptosis, modulating cell cycle arrest, and reducing mitotic slippage. The manuscript was written in a clear manner. The reviewer’s comments are listed below.

1) The authors only described the difference of Mcl-1 expression between non-muscle invasive bladder cancer and muscle invasive bladder cancer using 72 bladder cancer samples. The authors need to show the detail of patients’ clinical characteristic and the prognostic role of Mcl-1 in their bladder cancer patients.

2) The authors evaluated the combination effect of obatoclax with paclitaxel by their protocol of 1uM paclitaxel for 40hr, followed by 1uM obatoclax for 8h 0.1uM. How did the authors decide their concentration and their exposure hours? Could the regimen of obatoclax followed by paclitaxel work in the same way?

3) To strength their results, the authors need to perform in vivo study.

Author Response

We gratefully acknowledge the positive general comments made by the reviewers and their critiques and informed suggestions, which we believe have given the manuscript more clarity and strength. An explanation of the changes that have been made is given below in response to each of the reviewer’s comments. We believe our revisions have addressed the reviewer’s concerns adequately. We also thank the Editor for her work and the opportunity given to us to submitting this revised version of the manuscript. 

We attach the reviewer's comments in a PDF file.

Round  2

Reviewer 2 Report

The authors revise their manuscript correctly.